# Parameter Efficient Multimodal Transformers for Video Representation Learning

**Sangho Lee, Youngjae Yu, Gunhee Kim**
Seoul National University
`{sangho.lee,yj.yu}@vision.snu.ac.kr, gunhee@snu.ac.kr`

**Thomas Breuel, Jan Kautz**
NVIDIA Research
`{tbreuel,jkautz}@nvidia.com`

**Yale Song**
Microsoft Research
`yalesong@microsoft.com`

## ABSTRACT

The recent success of Transformers in the language domain has motivated adapting it to a multimodal setting, where a new visual model is trained in tandem with an already pretrained language model. However, due to the excessive memory requirements from Transformers, existing work typically fixes the language model and train only the vision module, which limits its ability to learn cross-modal information in an end-to-end manner. In this work, we focus on reducing the parameters of multimodal Transformers in the context of audio-visual video representation learning. We alleviate the high memory requirement by sharing the parameters of Transformers across layers and modalities; we decompose the Transformer into modality-specific and modality-shared parts so that the model learns the dynamics of each modality both individually and together, and propose a novel parameter sharing scheme based on low-rank approximation. We show that our approach reduces parameters of the Transformers up to 97%, allowing us to train our model end-to-end from scratch. We also propose a negative sampling approach based on an instance similarity measured on the CNN embedding space that our model learns together with the Transformers. To demonstrate our approach, we pretrain our model on 30-second clips (480 frames) from Kinetics-700 and transfer it to audio-visual classification tasks.

## 1 INTRODUCTION

Learning multimodal representation from unlabeled videos has received considerable attention (Baltrušaitis et al., 2018). Audio-visual learning is of particular interest due to the abundance of videos with natural audio-visual co-occurrence (Owens & Efros, 2018; Owens et al., 2018; Arandjelovic & Zisserman, 2018; Ephrat et al., 2018; Gao & Grauman, 2019; Alwassel et al., 2019). However, existing approaches learn localized representations from short videos (hundreds of milliseconds to just under a few seconds), capturing only *short-term* dependencies in data. While this is useful for certain applications, e.g., source separation (Ephrat et al., 2018) and atomic action recognition (Gu et al., 2018), learning representation that captures *long-term* dependencies is equally important, e.g., for activity recognition (Kay et al., 2017; Carreira et al., 2019; Sigurdsson et al., 2016). Unfortunately, processing long videos requires large memory resource and capturing long-term dependencies is a long-standing problem (Hochreiter & Schmidhuber, 1997; Cho et al., 2014; Vaswani et al., 2017).

In language understanding, strong progress has been made in large-scale learning of contextualized language representations using Transformers (Vaswani et al., 2017; Howard & Ruder, 2018; Peters et al., 2018; Radford et al., 2018; 2019; Devlin et al., 2019; Liu et al., 2019; Yang et al., 2019). Riding on the success of Transformers, several recent works have extended it to the multimodal setting by adding an additional vision module to the Transformer framework (Sun et al., 2019b; Lu et al., 2019). However, these models are typically not end-to-end trained; they rely on a language-pretrained BERT (Devlin et al., 2019), which is fixed throughout, and train only the visual components. While the pretrained BERT helps accelerate convergence and brings reliable extra supervision signal to the

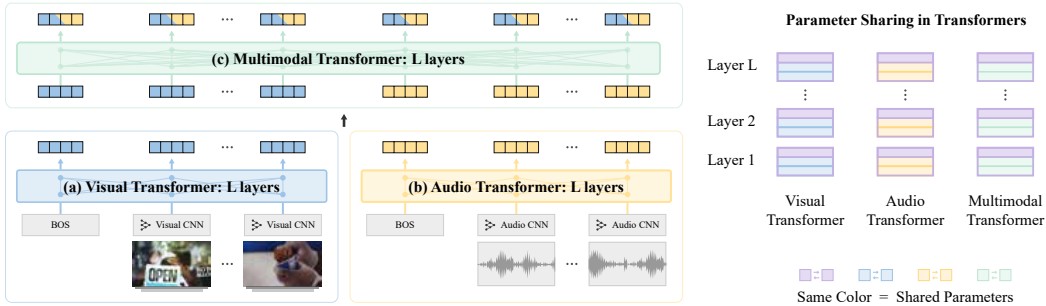

Figure 1: **(Left)** Our model consists of CNNs encoding short-term dynamics of each modality and Transformers encoding long-term dynamics of audio-visual information from videos. **(Right)** To alleviate excessive memory requirements, we propose an efficient parameter sharing scheme based on matrix decomposition with low-rank approximation, which allows us to train our model end-to-end.

vision component, this partial learning setup can be undesirable if the text data comes from different distributions (of topics, dialects, or foreign languages) or if we want to apply it to different modalities (e.g., audio-visual). Unfortunately, end-to-end training of such multimodal Transformer architectures is challenging for most existing compute environments due to the excessive memory requirement.

In this work, we make three key contributions. First, we propose an *end-to-end trainable* bidirectional transformer architecture that learns contextualized audio-visual representations of long videos. Our model, shown in Figure 1, consists of audio/visual CNNs, audio/visual Transformers, and a multi-modal Transformer. The CNNs operate on short (e.g., one second) video clips and are intended to capture short-term dynamics within each modality. The Transformer layers operate on long video sequences (e.g., 30 seconds), capturing long-term dynamics. To enable end-to-end training, we propose a novel parameter reduction technique that shares parts of weight parameters across Transformers and across layers within each Transformer. We show that this results in up to 97% parameter reduction, enabling end-to-end training of our model, with a minimal performance degradation. To the best of our knowledge, our work is the first to report end-to-end trained multimodal Transformers, and the first to apply Transformers for audio-visual representation learning.

The quality of negative samples is crucial in contrastive learning, which is part of our learning objective. As our second contribution, we propose a *content-aware* negative sampling strategy that favors negatives sufficiently similar to a positive instance. Our approach measures the similarity by reusing the CNN embeddings obtained during model training, and thus do not introduce extra parameters to learn. We show that this improves performance over the standard sampling strategies.

Our third contribution is a systematic evaluation of different modality fusion strategies. Existing works on multimodal BERT (all using vision-and-language data) typically apply one fusion strategy without thoroughly comparing with alternatives, e.g., some works perform early fusion (Sun et al., 2019b; Su et al., 2020) while others perform mid-level fusion (Lu et al., 2019; Tan & Bansal, 2019). As a result, it is unclear how different fusion methods affect the final performance. In this work, we compare three fusion strategies (early, mid, late) and show the superiority of mid-level fusion.

To demonstrate our approach, we pretrain our model on long (30-second) video clips from Kinetics-700 (Carreira et al., 2019) and finetune it on various video classification tasks. One benefit of the modular design of our architecture is flexibility: once pretrained, we can use any of the subnetworks for downstream tasks depending on the modalities involved (audio-only, visual-only, audio-visual) and video lengths (short and long). To show this, we evaluate our model on UCF101 (Soomro et al., 2012) and ESC-50 (Gemmeke et al., 2017) for short-term visual/audio classification, and Charades (Sigurdsson et al., 2016) and Kinetics-Sounds (Arandjelovic & Zisserman, 2017) for long-term audio-visual action recognition.

## 2 APPROACH

Figure 1 shows an overview of the proposed model architecture. The input to our model is a sequence of visual clips $\mathbf{v}_{1:T}$ and the corresponding sequence of audio streams $\mathbf{a}_{1:T}$. For example, each

sequence is a 30 second-long video divided into 30 non-overlapping clips (each clip is one second long). We divide our model into three parts with different characteristics, which are explained below.

**Local Feature Embedding.** We feed each of $T$ video clips to a visual CNN $f_V(\mathbf{v}_t)$ to obtain $\mathbf{x}_{1:T}^v \in \mathbb{R}^{T \times D}$, and each audio stream to an audio CNN $f_A(\mathbf{a}_t)$ to obtain $\mathbf{x}_{1:T}^a \in \mathbb{R}^{T \times D}$.[1] Intuitively, the CNN outputs are *temporally local* embeddings as they have access to only a short-range temporal window of the entire video sequence. Thus, they are suitable for representing short-range atomic actions (e.g., sit down, raise arms) that constitute long-range events (e.g., gym workout). We use the SlowFast network (Feichtenhofer et al., 2019) with a ResNet-50 backbone (He et al., 2016) as a visual CNN $f_V$, and a ResNet-50 as an audio CNN $f_A$. The weights of both CNNs are randomly initialized and trained end-to-end with the Transformer layers.

**Unimodal Contextualized Embedding.** The local feature embeddings capture short-term dynamics but lack long-term contextual information. We use Transformers (Vaswani et al., 2017) to enrich the embeddings with sequence-level context. We start by learning *unimodal* contextualized representations using the visual Transformer $g_V$ and the audio Transformer $g_A$, respectively.

The Transformer consists of L layers, each with two sub-layers: a multi-head attention layer and a feed-forward layer. Given an input sequence of embeddings $\mathbf{x} \in \mathbb{R}^{T \times D}$ and $A$ attention heads, the $j$-th head in the attention layer computes the output embedding sequence $\mathbf{a}_j \in \mathbb{R}^{T \times \gamma}, \gamma = D/A$ as

$$\mathbf{a}_j = \text{softmax}\left(\frac{Q_j K_j^\top}{\sqrt{\gamma}}\right) V_j, \qquad Q_j = \mathbf{x}W_j^q, K_j = \mathbf{x}W_j^k, V_j = \mathbf{x}W_j^v \tag{1}$$

where $W_j^q, W_j^k, W_j^v \in \mathbb{R}^{D \times \gamma}$ are weight matrices for computing the (`query`, `key`, `value`) triplet given the input $\mathbf{x}$. This operation is repeated for each attention head, and the outputs are combined (with concatenation followed by one linear layer with weights $W^b \in \mathbb{R}^{D \times D}$), producing $\mathbf{a} \in \mathbb{R}^{T \times D}$. Next, the feed-forward layer takes this intermediate output and computes $\mathbf{o} \in \mathbb{R}^{T \times D}$ using a two-layer fully-connected network with weights $W^c \in \mathbb{R}^{D \times E}$ and $W^d \in \mathbb{R}^{E \times D}$. The output of each sub-layer is computed using a residual function followed by layer normalization (Ba et al., 2016), i.e., $\text{LayerNorm}(x + \text{Sublayer}(x))$. In this work, we set the number of layers $L = 6$, the number of attention heads $A = 12$, the feature dimension $D = 768$ and the intermediate dimension $E = 3072$. For simplicity, we use this design for all layers across all three Transformers in our model.

Before feeding local embeddings $\mathbf{x}^v$ and $\mathbf{x}^a$ to unimodal Transformers, we augment them with "positional" embeddings. Specifically, we append to the beginning of each sequence a special vector `BOS` (beginning of sequence), i.e., $\mathbf{x}_0^v$ for visual and $\mathbf{x}_0^a$ for audio streams; their dimensions are same as $\mathbf{x}_t^v$ and $\mathbf{x}_t^a$, respectively. We also define positional embeddings $\mathbf{p}_{0:T}$ encoding time indices (we call this "time" embedding). This is necessary to preserve information about temporal ordering of local feature embeddings, which is otherwise lost in Eqn. 1. We combine them via layer normalization,

$$\mathbf{u}_t^v = \text{LayerNorm}(\mathbf{x}_t^v + \mathbf{p}_t^v), \quad \mathbf{u}_t^a = \text{LayerNorm}(\mathbf{x}_t^a + \mathbf{p}_t^a), \quad \forall t \in [0, T] \tag{2}$$

We initialize $\{\mathbf{x}_0^v, \mathbf{x}_0^a, \mathbf{p}_{0:T}^v, \mathbf{p}_{0:T}^a\}$ to the normal distribution and train them with the rest of the model.

We feed the augmented visual embeddings into the visual Transformer $g_V$ and obtain $\mathbf{y}_{0:T}^v = g_V(\mathbf{u}_{0:T}^v)$, and similarly obtain $\mathbf{y}_{0:T}^a = g_A(\mathbf{u}_{0:T}^a)$. The embeddings at each time step has a direct access to the entire input sequence regardless of their position (it has a one-step signal path during forward and backward inference). Multiple layers of such feature transformation thus allow the resulting embedding to be deeply contextualized in the time dimension. We denote the output embeddings corresponding to the `BOS` positions by $\text{BOS}_g^v = \mathbf{y}_0^v$ and $\text{BOS}_g^a = \mathbf{y}_0^a$, and designate them as the *summary* embeddings representing the sequence of each modality.

**Multimodal Contextualized Embedding.** The unimodal embeddings capture long-term temporal context but miss out on cross-modal information. The final step in forward inference is to use a multimodal Transformer $h_{AV}$ to obtain embeddings contextualized in the audio-visual space.

We first augment the embeddings $\mathbf{y}_{0:T}^v$ and $\mathbf{y}_{0:T}^a$ with modality and time embeddings. The modality embeddings $\mathbf{m}^v$ and $\mathbf{m}^a$ are vectors of the same dimension as $\mathbf{y}_t^v$ and $\mathbf{y}_t^a$, respectively. We share $\mathbf{m}^v$ (and $\mathbf{m}^a$) across all the unimodal embeddings $\mathbf{y}_{0:T}^v$ (and $\mathbf{y}_{0:T}^a$); thus, they add modality-discriminative information to the Transformer. We also add time embeddings $\mathbf{p}_{0:T}$ as before; however, unlike in

---

[1]For notational simplicity, we drop the subscripts to refer to the entire sequence unless distinction is necessary.

the previous step, we share the same $\mathbf{p}_{0:T}$ between embeddings from the two modalities to correctly indicate the time indices. We augment the modality and time embeddings via layer normalization,

$$\mathbf{w}_t^v = \text{LayerNorm}(\mathbf{y}_t^v + \mathbf{p}_t + \mathbf{m}^v), \ \mathbf{w}_t^a = \text{LayerNorm}(\mathbf{y}_t^a + \mathbf{p}_t + \mathbf{m}^a), \ \forall t \in [0, T] \quad (3)$$

We feed the augmented visual embeddings $\mathbf{w}_{0:T}^v$ and audio embeddings $\mathbf{w}_{0:T}^a$ to the multimodal Transformer $h_{AV}$, one after another, and obtain $\mathbf{z}_{0:(2T+1)} = h_{AV}([\mathbf{w}_{0:T}^v; \mathbf{w}_{0:T}^a])$. We again denote the output embeddings corresponding to the BOS positions by $\text{BOS}_h^v = \mathbf{z}_0^v (= \mathbf{z}_0)$ and $\text{BOS}_h^a = \mathbf{z}_0^a (= \mathbf{z}_{T+1})$, and use them as summary embeddings encoding multimodal context.

We emphasize the importance of feeding $\mathbf{w}_{0:T}^v$ and $\mathbf{w}_{0:T}^a$ one after another. An alternative would be concatenating them before feeding them to $h_{AV}$ and obtaining an output $\mathbf{z}_{0:T}$ (instead of $\mathbf{z}_{0:(2T+1)}$). However, this restricts the Transformer to access audio-visual embeddings only from the same time slices, which could be problematic when there is a temporally asynchronous relationship between the two modalities (e.g., a visual clip matches with sound captured a few times steps before) (Kazakos et al., 2019; Morgado et al., 2020). By arranging the two sequences one after the other, the Transformer can mix-and-match appropriate audio-visual embeddings in an asynchronous manner. Another practical concern with the alternative approach is that it significantly increases the model size; the weight matrices $W_q, W_k, W_v$ grow quadratically with the input feature dimension $D$. Serializing the input resolves both issues.

## 2.1 SELF-SUPERVISED PRETRAINING OBJECTIVES

**Task 1: Masked Embedding Prediction (MEP).** BERT (Devlin et al., 2019) is trained using the masked language model (MLM) task, which randomly selects input tokens and replaces them with a mask token. The model is then trained to predict the original (unmasked) tokens by solving a classification task with a cross-entropy loss. However, inputs to our model are real-valued audio-visual signals (rather than discrete tokens),[2] so applying the MLM task requires input discretization, which causes information loss (Lu et al., 2019; Sun et al., 2019a). We instead train our model to identify the correct visual clip or audio stream compared to a set of negative samples in a contrastive manner, which does not require input discretization.

We formulate our MEP task using InfoNCE (Oord et al., 2018), which is the softmax version of the noise contrastive estimation (NCE) (Gutmann & Hyvärinen, 2010). Let $\tilde{\mathbf{o}}_t$ be the $t$-th output of any of the three Transformers obtained by masking the $t$-th input $\mathbf{x}_t$. Our InfoNCE loss is then defined as

$$\mathcal{L}_{\text{NCE}}(\mathbf{x}, \tilde{\mathbf{o}}) = -\mathbb{E}_{\mathbf{x}} \left[ \sum_t \log \frac{\text{I}(\mathbf{x}_t, \tilde{\mathbf{o}}_t)}{\text{I}(\mathbf{x}_t, \tilde{\mathbf{o}}_t) + \sum_{j \in \text{neg}(t)} \text{I}(\mathbf{x}_j, \tilde{\mathbf{o}}_t)} \right], \quad (4)$$

where $\text{neg}(t)$ are negative sample indices and the compatibility function $\text{I}(\mathbf{x}_t, \tilde{\mathbf{o}}_t)$ is,

$$\text{I}(\mathbf{x}_t, \tilde{\mathbf{o}}_t) = \exp \left( \text{FFN}^\top(\tilde{\mathbf{o}}_t) W_I \mathbf{x}_t \right), \quad (5)$$

where $W_I \in \mathbb{R}^{P \times D}$ ($P = 256$) and FFN is a two-layer feed-forward network. The use of a non-linear prediction head has shown to improve the quality of the representations learned in a contrastive learning setup (Chen et al., 2020); following the recent work in Transformers (Devlin et al., 2019; Liu et al., 2019; Lan et al., 2020), we use a GELU non-linear activation function (Hendrycks & Gimpel, 2016) in FFN. Optimizing Eqn. 4 enforces $\text{I}(\mathbf{x}_t, \tilde{\mathbf{o}}_t)$ to approximate the density ratio $\frac{p(\mathbf{x}_t | \tilde{\mathbf{o}}_t)}{p(\mathbf{x}_t)}$; this can be seen as maximizing the mutual information between $\mathbf{x}_t$ and $\tilde{\mathbf{o}}_t$ (Oord et al., 2018). Intuitively, this encourages the Transformer to capture the underlying dynamics of $\mathbf{x}$ from each modality without explicitly learning a generative model $p(\mathbf{x}_t | \tilde{\mathbf{o}}_t)$.

**Negative sampling.** We find that a good negative sampling strategy is essential for the model's convergence. Existing approaches either use all but $\mathbf{x}_t$ (positive) within a mini-batch as negative samples or limit it to the current sequence only. However, both these methods ignore the data content and thus can miss useful negatives. Oord et al. (2018) showed that leveraging prior knowledge about data can improve the negative sample quality (e.g., by sampling negatives from the same speaker as the positive). Unfortunately, such prior knowledge is often not available in unlabeled videos.

---

[2]In the form of RGB images and log-mel-scaled spectrograms.

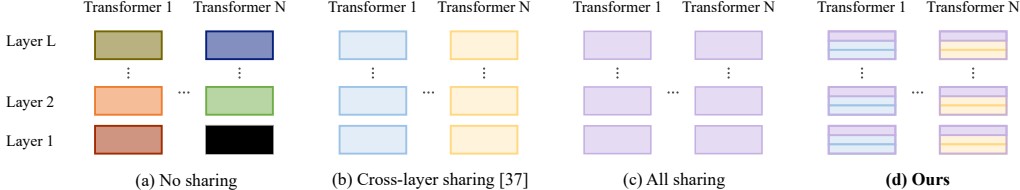

Figure 2: Comparison of parameter sharing schemes. Ours combines (b) and (c) but decomposes weights in each layer into private and shared parts so only the latter is shared across Transformers.

We propose a *content-aware* negative sampling strategy that favors negatives sufficiently *similar* to a positive instance in the CNN embedding space; we call our approach `CANS-Similar`. Our approach is inspired by Ulyanov et al. (2018) who showed that randomly initialized CNNs provide a strong prior over natural images due to the inductive bias already built into the design of the CNNs. This suggests that our local feature embeddings $\mathbf{x}^v$ (and $\mathbf{x}^a$) can capture the underlying statistical regularities in video clips (and audio streams) right from the beginning, which can be sufficient to assess the similarity/dissimilarity between clips. Therefore, the distance measured on them can approximate content dissimilarity well (and this will improve as the training progresses).

Motivated by this, we sample the negatives based on local feature embeddings $\mathbf{x}^v$ (and $\mathbf{x}^a$). Specifically, we compute a pairwise $\ell_2$ distance between $\mathbf{x}_t$ (positive) and all other instances within a mini-batch, and normalize them to the [0, 1] interval. To remove samples that are either too similar or too different from the positive sample, we discard instances that fall outside the 95% confidence interval in the normalized distance space. We then sample the negatives from the remainder using the normalized distance as sampling probability. This makes instances similar to the positive instance have more chance to become negatives. We emphasize the importance of sampling, instead of deterministically taking top most similar samples; the stochasticity allows our model to be robust to potentially inaccurate distance estimates because samples with low probabilities will still have a chance to be selected as negatives.

Finally, our MEP loss is the InfoNCE loss computed on all three Transformers,

$$\mathcal{L}_{\text{MEP}} = \mathcal{L}_{\text{NCE}}(\mathbf{x}^a, \tilde{\mathbf{y}}^a) + \mathcal{L}_{\text{NCE}}(\mathbf{x}^v, \tilde{\mathbf{y}}^v) + \mathcal{L}_{\text{NCE}}([\mathbf{x}^a; \mathbf{x}^v], \tilde{\mathbf{z}}) \quad (6)$$

**Task 2: Correct Pair Prediction (CPP).** The MEP task encourages our model to learn the underlying dynamics within each modality. To help our model learn cross-modal dynamics, we design a task that predicts whether a pair of audio-visual embeddings is from the same video. Specifically, we define two binary classifiers, one for the two unimodal Transformers and another for the multimodal Transformer. Each classifier takes as input either $\mathbf{s}_g = [\mathbf{y}_0^v; \mathbf{y}_0^a]$ (and $[\mathbf{z}_0^v; \mathbf{z}_0^a]$), a pair of audio-visual "summary" embeddings corresponding to the `BOS` positions, or $\mathbf{s}_h = [\mathbf{y}_t^v; \mathbf{y}_t^a]$ (or $[\mathbf{z}_t^v; \mathbf{z}_t^a]$), the output embeddings sampled at random positions (we take two random positions $t \in [1, T]$). The classifier predicts $p(c|\mathbf{s})$ indicating whether the pair is from the same video ($c = 1$) or from different videos ($c = 0$). We train the classifiers with a binary cross-entropy loss,

$$\mathcal{L}_{CPP} = -\mathbb{E}_{\mathbf{x},\mathbf{y}} \left[ c \cdot \log p(c|\mathbf{s}_g) + c \cdot \log p(c|\mathbf{s}_h) \right] \quad (7)$$

where $\cdot$ is the inner product. We generate a random derangement of the input mini-batch so that the number of positive and negative pairs are guaranteed to be the same.

**Overall Pretraining Objective**. We train our model end-to-end from scratch by optimizing $\mathcal{L}_{MEP} + \alpha \mathcal{L}_{CPP}$ with a balancing term $\alpha$. We find our model is insensitive to this term, so we set $\alpha = 1.0$.

## 2.2 PARAMETER REDUCTION

Optimizing our model is challenging due to the large memory requirement. The most expensive part is the Transformers, which take up 82% of model parameters. One could reduce the model size by making the Transformers shallower, but the depth of Transformers has shown to be crucial to get good performance (Devlin et al., 2019). We propose to reduce the model size by aggressively sharing parts of weights across Transformers as well as layers within each Transformer (see Figure 2 (d)).

**Sharing across Transformers.** We first consider sharing weights across Transformers. Each Transformer encodes data coming from different distributions: $g_V$ encodes $\mathbf{x}^v$, $g_A$ encodes $\mathbf{x}^a$, and

$h_{AV}$ encodes $(\mathbf{y}^v, \mathbf{y}^a)$. These input distributions may each exhibit different dynamics, yet together share certain regularities because they all come from the same videos. Motivated by this, we decompose Transformer weights into shared and private parts so that different patterns can be learned in a parameter-efficient manner. Recall that each layer of a Transformer contains weights $\{W^q, W^k, W^v, W^b, W^c, W^d\}$. We decompose each of these weights into $W = U\Sigma V^\top$, where $W \in \mathbb{R}^{M \times N}, U \in \mathbb{R}^{M \times O}, \Sigma \in \mathbb{R}^{O \times O}, V \in \mathbb{R}^{N \times O}$. We perform low-rank approximation of $W$ by setting the rank $O \ll M, N$, and share $U$ across Transformers while keeping $\Sigma$ and $V$ private to each Transformer. This helps reduce parameters because $MO + 3(O^2 + NO) \ll 3MN$. We experimented with different matrix ranks $O$ but the differences were small; we set $O = 128$ ($M, N = 768$ or $3072$).

The decomposition converts a linear projection of input $W\mathbf{x}$ into a series of (unconstrained) linear projections $U\Sigma V^\top \mathbf{x}$. However, this can cause numerical instability during optimization (Nocedal & Wright, 2006). We could perform the Singular Value Decomposition (SVD) over $W$ so that it performs rotation ($V^\top$), stretch ($\Sigma$), and rotation ($U$) with orthogonal basis vectors in $U$ and $V$. Unfortunately, solving the full SVD has a computational complexity of $\mathcal{O}(\max(M, N)^2)$ (Golub & Van Loan, 2012). Here, we put an orthogonality constraint only on $\Sigma$ and perform projection ($V^\top$), rotation ($\Sigma$), and projection ($U$) of input $\mathbf{x}$. In addition, we put $V^\top \mathbf{x}$ in a unit sphere (via $\ell_2$-normalization) before rotating it with $\Sigma$. This not only improves numerical stability, but also removes magnitude information in $V^\top \mathbf{x}$ and keeps angular information only, which has been shown to provide sample discriminative information (Chen et al., 2019a). To impose the orthogonality constraint on $\Sigma$, we use the Padé approximation with a scale-squaring trick of Lezcano-Casado & Martínez-Rubio (2019). Intuitively, we linearly project $\mathbf{x}$ onto a unit sphere ($V^\top \mathbf{x}$) and rotate it ($\Sigma V^\top \mathbf{x}$) in each Transformer so that it captures the dynamics of each input distribution independently. We then project it to the shared space via $U$, capturing shared regularities across all three Transformers.

**Sharing across Layers.** Recently, Bai et al. (2019a) showed that sharing parameters across layers in deep neural networks does not hurt the representational power of the network. Furthermore, (Lan et al., 2020) demonstrated that cross-layer parameter sharing in the Transformer leads to a lighter and faster-to-train model without sacrificing the performance on various language understanding benchmarks. Motivated by this, we let each Transformer share parameters across different layers.

## 3 EXPERIMENTS

We pretrain our model on Kinetics-700 (Carreira et al., 2019) or AudioSet (Gemmeke et al., 2017) and finetune it on various downstream tasks. The official release of Kinetics-700 contains 10-second clips only, so we download 410K original videos from YouTube and take 30-second clips from each video. For fair comparison with prior work, we use 10-second clips from the official release of AudioSet (we used 1.8M clips). We pretrain our model on 64 NVIDIA Tesla V100 GPUs with a batch size of 256 for 220K iterations. For downstream tasks, we evaluate on *short*-video/audio classification using UCF-101 (Soomro et al., 2012) (13K clips from 101 classes; 7.2 seconds on average) and ESC-50 (Gemmeke et al., 2017) (2K clips from 50 classes; 5 seconds), and on *long*-video classification using Kinetics-Sounds (Arandjelovic & Zisserman, 2017) (23K videos from 32 classes; 10 seconds on average) and Charades (Sigurdsson et al., 2016) (10K videos from 157 classes; 30 seconds on average). We describe various details about experimental setup in Appendix.

### 3.1 RESULTS AND DISCUSSION

**Multimodal Fusion Methods.** To evaluate different fusion methods on the quality of learned representation, we test the following settings: (i) `Early` uses a single multimodal Transformer with $2 \times L$ layers, (ii) `Mid` is our approach described in Figure 1, (iii) `Late` uses two unimodal Transformers each with $2 \times L$ layers. All the methods are pretrained on audio-visual data using CPP and MEP losses, except for (iv) `Late-w/o-CPP` where we use only the MEP loss. We finetune the pretrained models on audio-visual, audio-only, and visual-only scenarios. For fair comparisons across different fusion methods, we do not perform parameter sharing in this ablation setting.

Table 1 (a) shows that `Early` and `Mid` outperform `Late` on the audio-visual scenario. This suggests the importance of encoding cross-modal information. Note that `Late-w/-CPP` gets cross-modal self-supervision, which gives marginal performance improvement over `Late-w/o-CPP`; however, both methods miss the opportunity to *encode* any cross-modal relationship, leading to inferior results.

| a) Fusion Method | Audio-Visual | Audio-only | Visual-only |
|---|---|---|---|
| Early | 64.9 / 89.8 | - / - | - / - |
| Late-w/-CPP | 61.0 / 88.7 | 52.3 / 80.8 | 41.0 / 71.3 |
| Late-w/o-CPP | 60.6 / 87.6 | 50.5 / 79.9 | 40.7 / 71.7 |
| Mid† | **65.7 / 89.9** | **53.5 / 82.7** | **42.5 / 73.2** |

| b) Sampling Method | top-1 | top-5 |
|---|---|---|
| Current-Sequence | 64.6 | 89.8 |
| Current-MiniBatch | 65.5 | 90.8 |
| CANS-Dissimilar | 66.2 | 91.1 |
| CANS-Similar† | **67.5** | **92.3** |

| c) Model | X.-L | X.-T | Params | top-1/5 |
|---|---|---|---|---|
| Multi-2 | ✗ | ✗ | 7M | 60.3 / 88.9 |
| Multi-6 | ✓ | ✗ | 21M | 65.7 / 89.9 |
| Multi-6 | ✓ | ✓(All) | 7M | 67.1 / 92.3 |
| Multi-6 | ✓ | ✓(Part†) | **4M** | **67.5 / 92.3** |

| d) Model | X.-L | X.-T | Params | top-1/5 |
|---|---|---|---|---|
| Vis-2 | ✗ | ✗ | 14M | 41.4 / 71.0 |
| Vis-2 | ✓ | ✗ | 7M | 41.2 / 72.9 |
| Vis-6 | ✗ | ✗ | 43M | 43.8 / 74.2 |
| Vis-6 | ✓ | ✗ | 7M | 43.5 / 73.7 |

Table 1: Ablation study on Kinetics-Sounds comparing: **(a; top-left)** multimodal fusion methods, **(b; top-right)** negative sampling strategies, and **(c & d; bottom)** parameter sharing schemes. X.-L: Cross-layer, X.-T: Cross-Transformer sharing. We report top-1 and top-5 accuracy (%). †: Ours.

While both Early and Late perform similarly in the audio-visual scenario, only Late can be used in unimodal downstream scenarios (c.f., Early requires the presence of both modalities). This has practical implications: Mid and Late can effectively handle missing modalities, i.e., once pretrained on audio-visual data, we can use it on any of audio-visual, audio-only, and visual-only scenarios. Our Mid fusion approach enjoys both the advantages, i.e., learning cross-modal relationship and being robust to missing modalities, achieving overall the best performance.

**Negative Sampling Strategies.** We compare four strategies: (i) Current-Sequence takes all but the positive instance from the same sequence as negatives, (ii) Current-MiniBatch takes all but the positive instance in the mini-batch as negatives; this subsumes Current-Sequence, (iii) CANS-Dissimilar stochastically samples negatives using a modified version of our content-aware negative sampling (CANS) that favors *dissimilar* samples, and (iv) CANS-Similar is our proposed CANS approach that favors negatives that are *similar* to the positive instance.

Table 1 (b) shows Current-Sequence is the least effective: It makes MEP too difficult because negatives are (sometimes too much) similar to positives. As a result, the training dynamics is dominated by CPP, which is relatively easier, leading to inferior performance. We make quite the contrary observations from Current-MiniBatch: the inclusion of negatives from different videos makes MEP easier and thus makes it dominate the training dynamics. Our CANS approach solves both these issues by eliminating negatives that are either almost identical to or trivial to distinguish from the positives, based on the 95% CI over the CNN embedding distances. It also samples negatives in a stochastic manner so a wide variety of samples can be included as negatives. Our proposed CANS-Similar can be considered as a "softened" version of Current-Sequence; it samples negatives that are similar to positives with a high probability (this can be considered as online hard negative mining), but it also takes instances from different videos with a lower probability. This balances out hard and easy negatives, making the MEP task effective.

**Parameter Sharing Schemes.** Our parameter reduction scheme reduces the number of parameters from 128M to 4M (by 97%) (Table 1 (c)). We reduce the model size by sharing weights across Transformers and across layers. We validate these ideas in two sets of experiments. Table 1 (c) compares cross-Transformer weight sharing schemes. We use Multi-6 that uses all three Transformers with 6 layers each, and compare four methods that correspond to Figure 2 (a)-(d). Note that No sharing is too large to fit in a Tesla V100 GPU (16GB) even with 2 samples, so we define Multi-2 that uses three Transformers with 2 layers each, and with the reduced number of attention heads $A$ to 5, the feature dimension $D$ to 320 and the intermediate dimension $E$ to 1280. We see that our proposed approach, Part, achieves the best performance with the least number of parameters. One might ask how Part leads to a smaller model when All shares all the weights across Transformers: We decompose weights $W = U\Sigma V^\top$ with low-rank approximation and share only $U$ across Transformers, while the $\Sigma V^\top$ part learns modality-specific dynamics. Table 1 (d) compares cross-layer weight sharing schemes using the visual Transformer with either 2 (Vis-2) or 6 (Vis-6) layers. The results show that sharing weights across layers does not hurt the performance, confirming the observations by Lan et al. (2020) in the audio-visual setting.

**Pretraining Objectives.** To evaluate the importance of MEP and CPP tasks, we test two settings: (i) Mid-w/o-CPP and (ii) Mid-w/o-MEP. On Kinetics-Sounds, these achieve 65.9% and 64.6%, respectively; ours achieve 67.5% (top-1 accuracy). The result show that the MEP task plays an important role during pretraining, confirming the findings from Sun et al. (2019a) that the InfoNCE

| a) Model | Net | Data | UCF |     | b) Model | Net | Data | ESC |     | c) Model | Charades | KS |
|---|---|---|---|---|---|---|---|---|---|---|---|---|
| ST-Puzzle | 3D-R18 | K400 | 65.8 | | SVM | MLP | - | 39.6 | | Random | 5.9 | - / - |
| ClipOrder | R(2+1)D | UCF | 72.4 | | ConvAE | CNN-4 | | 39.9 | | ATF | 18.3 | - / - |
| DPC | 3D-R34 | K400 | 75.7 | | RF | MLP | - | 44.3 | | ATF (OF) | 22.4 | - / - |
| CBT | S3D | K600 | 79.5 | | ConvNet | CNN-4 | | 64.5 | | V-CNN | 18.7 | 45.8 / 73.3 |
| MultiSens | 3D-R18 | AS | 82.1 | | SoundNet | CNN-8 | FS | 74.2 | | A-CNN | 18.9 | 49.4 / 76.9 |
| AVTS | MC3-18 | K400 | 85.8 | | $L^3$-Net | CNN-8 | FS | 79.3 | | M-CNN | 23.1 | 59.4 / 83.6 |
| AVTS | MC3-18 | AS | **89.0** | | DMC | VGG-ish | FS | 79.8 | | V-BERT | 26.0 | 49.5 / 78.9 |
| V-CNN[†] | SlowFast | K700 | 85.2 | | AVTS | VGG-M | AS | 80.6 | | A-BERT | 27.4 | 58.9 / 85.7 |
| V-CNN[†] | SlowFast | AS | 86.1 | | A-CNN[†] | R50 | AS | **81.5** | | M-BERT[†] | **29.5** | **75.6 / 94.6** |

**Datasets**. K: Kinetics, AS: AudioSet, FS: Flicker-SoundNet, KS: Kinetics-Sounds. **Baselines**. ST-Puzzle (Kim et al., 2019), ClipOrder (Xu et al., 2019), DPC (Han et al., 2019), CBT (Sun et al., 2019a), MultiSens (Owens & Efros, 2018), AVTS (Korbar et al., 2018), AE (Aytar et al., 2016), SVM (Piczak, 2015a), RF (Piczak, 2015a), ConvNet (Piczak, 2015b), SoundNet (Aytar et al., 2016), $L^3$-Net (Arandjelovic & Zisserman, 2017), DMC (Hu et al., 2019), ATF (Sigurdsson et al., 2017)

Table 2: **(a; left):** Short video classification results on UCF101 (mean accuracy (%)). **(b; center):** Short audio classification results on ESC-50 (mean accuracy (%)). **(c; right):** Long video classification results on Charades (mAP) and Kinetics-Sounds (KS; top-1/5 accuracy (%)). [†]: Ours.

loss, as deployed in CBT, is effective in the cross-modal setting. The result also shows that augmenting MEP with CPP provides further performance improvement by learning cross-modal correspondence.

**Downstream Evaluation.** We pretrain our model with `Mid` fusion using MEP and CPP tasks (with `CANS-Similar`), and employ `Part` weight sharing. We use either Kinetics-700 or AudioSet for fair comparisons with prior work. Table 2 (a)/(b) shows *short*-video/audio classification results on UCF-101/ESC-50. For fair comparisons to the baselines, we use only the visual/audio CNN (no Transformers); we finetune a linear classifier on top of the visual CNN end-to-end for UCF-101, and train a multi-class one-vs-all linear SVM on top of the fixed audio CNN for ESC-50. Although our model is pretrained on long video clips with no direct supervision to the CNN layers (gradients must flow *through* Transformers), it outperforms most of the baselines (except for AVTS on UCF-101) that received direct supervision from short video clips. We note that, similar to ours, CBT (Su et al., 2020) is a multimodal Transformer pretrained on long video clips and thus is the most meaningful comparison to ours; ours outperform CBT on UCF-101 by 5.7%. For sound classification, our approach outperform all existing published results.

Table 2 (c) shows *long*-video classification results on Charades and Kinetics-Sounds (KS) when pretrained on Kinetics-700. We test `Visual-only (V)`, `Audio-only (A)`, and `Multimodal (M)` settings to verify the benefit of multimodal learning. Because there is no published self-supervised learning results on these datasets, we demonstrate long-term representations by comparing CNNs (`CNN`; short-term) to Transformers (`BERT`; long-term) on KS that contains 10-second clips. Since CNNs process 1-second clips, we feed 10 non-overlapping clips to CNNs and average the prediction output. In all settings, we add a 2-layer MLP with softmax classifier on top. The results show that Transformers outperform CNNs on Kinetics-Sounds, suggesting the superiority of long-term representations. We also see that combining audio-visual information performs the best. We notice that audio representations are generally stronger than visual representations; we believe that learning discriminative visual representations is generally more challenging, especially when the CNNs receive (self-)supervision signals *only indirectly* through Transformers. We believe that providing (self-)supervision directly to CNNs, e.g., by first pretraining CNNs on 3D rotation prediction (Jing et al., 2018) and then jointly training the whole model (as was done in CBT (Sun et al., 2019a)), could further improve performance. Incorporating contrastive learning (Chen et al., 2020) over the CNN embeddings and training the whole model end-to-end is another promising direction for future work.

## 4 RELATED WORK

**Multimodal BERT.** Extending BERT (Devlin et al., 2019) to vision-and-language has been actively studied. Existing work typically adopt early fusion (Li et al., 2019; Alberti et al., 2019; Sun et al., 2019b; Li et al., 2020; Zhou et al., 2020; Su et al., 2020; Chen et al., 2019b; Zhu & Yang, 2020) or mid fusion (Tan & Bansal, 2019; Lu et al., 2019; Sun et al., 2019a; Luo et al., 2020) without thorough validation, and they train only visual components while relying on a language-pretrained BERT. Although there have been some efforts to leverage the Transformer architecture (Vaswani et al., 2017) for audio and visual inputs (Boes & Van hamme, 2019; Tian et al., 2020), our approach is the first to demonstrate multimodal audio-visual BERT trained from scratch in an end-to-end manner. This is enabled by our novel parameter reduction technique, which is one of our main technical contributions.

**Audio-Visual Learning.** Early work in audio-visual learning focused on speech signals, improving audio-visual speech recognition than unimodal approaches (Ngiam et al., 2011; Srivastava & Salakhutdinov, 2012). Recent approaches leverage unlabeled videos from specific domains (Owens et al., 2016; Gao & Grauman, 2019; Zhao et al., 2018; Ephrat et al., 2018; Alwassel et al., 2019; Miech et al., 2020; Piergiovanni et al., 2020) and often demonstrate on audio-visual source separation, localization, and co-segmentation. However, these approaches rely on short-term audio-visual correspondence and thus may not generalize to long-term video recognition that requires global context (as was suggested in (Hjelm et al., 2019)), which this work focuses on.

**Parameter Reduction.** Network pruning (Reed, 1993; Caron et al., 2020) trains a large model and then reduces its size while maintaining performance. Reducing the size of CNNs for mobile applications is an active research area (Rastegari et al., 2016; Howard et al., 2017; 2019; Zhang et al., 2018; Iandola et al., 2016). Our work is closely related to the work that shares parameters across layers in deep neural networks. Trellis network (Bai et al., 2019b) is a temporal convolutional architecture with weight-tying across time and depth. Similar to ours, Universal Transformer (Dehghani et al., 2019), RSNMT (Dabre & Fujita, 2019), DEQ (Bai et al., 2019a), ALBERT (Lan et al., 2020) share weights across layers in Transformers. We combine this idea with our novel cross-Transformer weight sharing, which decomposes weight matrices with low-rank approximation.

**Negative Sampling.** Hard negative mining has been shown to be crucial for contrastive learning (Arandjelovic & Zisserman, 2017; Owens & Efros, 2018; Korbar et al., 2018; Schroff et al., 2015; Zhuang et al., 2019; Morgado et al., 2020; Wu et al., 2020). Korbar et al. (2018) use the time difference between clips to approximate clip similarity (i.e., clips that are further apart are deemed more different). However, such an assumption may not hold for real-world videos, e.g., periodic actions such as push-ups. Unlike this line of approaches, we directly use the feature embeddings learned by our model. Several apparoaches adapted a similar idea (Schroff et al., 2015; Zhuang et al., 2019; Morgado et al., 2020; Wu et al., 2020). Different from prior work, we bring the stochasticity to the sampling procedure by using the content similarity as the sampling probability; this helps reduce potential errors especially during the early stage of training.

## 5  CONCLUSION

We introduced a multimodal bidirectional Transformer architecture for self-supervised learning of contextualized audio-visual representation from unlabeled videos. Our main technical contributions include: (1) we propose a parameter efficient multimodal Transformers based on matrix decomposition with low-rank approximation; (2) we propose a novel content-aware negative sampling technique for contrastive learning. We demonstrate a successful end-to-end training of multimodal Transformers for audio-visual learning (which is, to the best of our knowledge, the first time in the literature). We also report comprehensive evaluation of various design decisions in multimodal learning.

**Acknowledgements.** This work was partially supported by Institute of Information & communications Technology Planning & Evaluation (IITP) grant funded by the Korea government (MSIT) (No.2017-0-01772, Video Turing Test, No.2019-0-01082, SW StarLab) and the international cooperation program by the NRF of Korea (NRF-2018K2A9A2A11080927).

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

## A   IMPLEMENTATION DETAILS

### A.1   ARCHITECTURES OF VISUAL/AUDIO CNNS

Table 3 shows the architectures of visual and audio CNNs we use for our model. For the visual CNN, we use the SlowFast network (Feichtenhofer et al., 2019) with a ResNet-50 backbone (He et al., 2016). We use the speed ratio $\alpha = 8$ and the channel ratio $\beta = 1/8$ for the SlowFast architecture, so $T_f = 8 \times T_s$. We use different values of $T_s$ and $T_f$ for different tasks. During pretraining, we set $T_s = 4$ and $T_f = 32$. During finetuning, we use $T_s = 8$ and $T_f = 64$ for short-video action classification on UCF101 (Soomro et al., 2012) while we use $T_s = 4$ and $T_f = 32$ for long-video action classification on Charades (Sigurdsson et al., 2016) and Kinetics-Sounds (Arandjelovic & Zisserman, 2017). For the audio CNN, we use a ResNet-50 without the downsampling layer $pool_1$ to preserve information along both frequency and time axis in early stages. We use different values of $T_a$ for different training phases. We set $T_a = 220$ for one-second clip during pretraining while we use $T_a = 440$ for two-second clip during finetuning.

| Stage | Visual CNN | | Audio CNN |
|---|---|---|---|
| | *Slow* pathway | *Fast* pathway | |
| raw clip | $3 \times T_s \times 112^2$ | $3 \times T_f \times 112^2$ | $128 \times T_a$ |
| $conv_1$ | $1 \times 7^2, 64$ 
 stride $1, 2^2$ | $5 \times 7^2, 8$ 
 stride $1, 2^2$ | $9 \times 9, 32$ 
 stride $1, 1$ |
| $pool_1$ | $1 \times 3^2, \text{max}$ 
 stride $1, 2^2$ | $1 \times 3^2, \text{max}$ 
 stride $1, 2^2$ | – |
| $res_2$ | $\begin{bmatrix} 1 \times 1^2, 64 \\ 1 \times 3^2, 64 \\ 1 \times 1^2, 256 \end{bmatrix} \times 3$ | $\begin{bmatrix} 3 \times 1^2, 8 \\ 1 \times 3^2, 8 \\ 1 \times 1^2, 32 \end{bmatrix} \times 3$ | $\begin{bmatrix} 1 \times 1, 32 \\ 3 \times 3, 32 \\ 1 \times 1, 128 \end{bmatrix} \times 3$ |
| $res_3$ | $\begin{bmatrix} 1 \times 1^2, 128 \\ 1 \times 3^2, 128 \\ 1 \times 1^2, 512 \end{bmatrix} \times 4$ | $\begin{bmatrix} 3 \times 1^2, 16 \\ 1 \times 3^2, 16 \\ 1 \times 1^2, 64 \end{bmatrix} \times 4$ | $\begin{bmatrix} 1 \times 1, 64 \\ 3 \times 3, 64 \\ 1 \times 1, 256 \end{bmatrix} \times 4$ |
| $res_4$ | $\begin{bmatrix} 3 \times 1^2, 256 \\ 1 \times 3^2, 256 \\ 1 \times 1^2, 1024 \end{bmatrix} \times 6$ | $\begin{bmatrix} 3 \times 1^2, 32 \\ 1 \times 3^2, 32 \\ 1 \times 1^2, 128 \end{bmatrix} \times 6$ | $\begin{bmatrix} 1 \times 1, 128 \\ 3 \times 3, 128 \\ 1 \times 1, 512 \end{bmatrix} \times 6$ |
| $res_5$ | $\begin{bmatrix} 3 \times 1^2, 512 \\ 1 \times 3^2, 512 \\ 1 \times 1^2, 2048 \end{bmatrix} \times 3$ | $\begin{bmatrix} 3 \times 1^2, 64 \\ 1 \times 3^2, 64 \\ 1 \times 1^2, 256 \end{bmatrix} \times 3$ | $\begin{bmatrix} 1 \times 1, 256 \\ 3 \times 3, 256 \\ 1 \times 1, 1024 \end{bmatrix} \times 3$ |

Table 3: The architectures of visual and audio CNNs. For the visual CNN, the input dimensions are denoted by *{channel size, temporal size, spatial size$^2$}*, kernels are denoted by *{temporal size, spatial size$^2$, channel size}* and strides are denoted by *{temporal stride, spatial stride$^2$}*. For the audio CNN, the input dimensions are denoted by *{frequency size, temporal size}*, kernels are denoted by *{frequency size, time size, channel size}* and strides are denoted by *{frequency stride, temporal stride}*.

### A.2   DATA PREPROCESSING

We preprocess the data by dividing $T$-second clips into $T$ non-overlapping parts ($T = 30$ for Kinetics-700 (Carreira et al., 2019) and $T = 10$ for AudioSet (Gemmeke et al., 2017)) and sampling 16 frames from each. For audio stream, we take waveform sampled at 44.1 kHz and convert it to log-mel-scaled spectrogram. We augment audio data with random frequency/time masking using SpecAugment (Park et al., 2019), and visual data with color normalization, random resizing, random horizontal flip, and

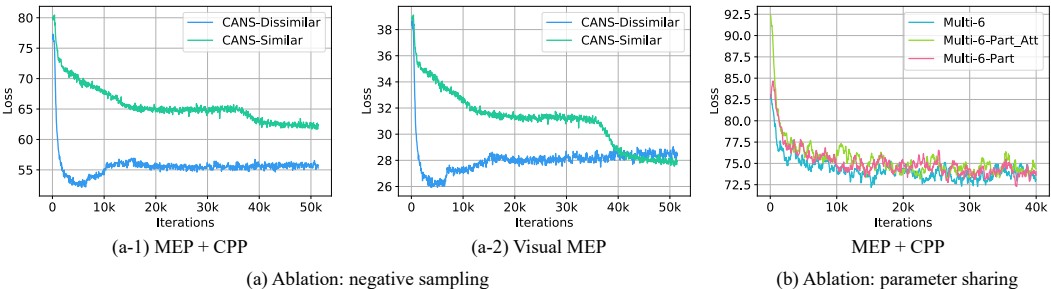

(a-1) MEP + CPP  (a-2) Visual MEP  MEP + CPP

(a) Ablation: negative sampling  (b) Ablation: parameter sharing

Figure 3: Loss curves during pretraining under different ablative settings. **(a)** compares Content-Aware Negative Sampling (CANS) that favors negatives that are dissimilar vs. similar to the positive instance. **(b)** compares different cross-Transformer weight sharing schemes; see the text for details.

random cropping to obtain $112 \times 112$ pixel frames; for test data, we resize videos to 128 pixels on the shorter side and take three equidistant crops of $128 \times 128$ pixels to cover the entire region. We also apply audio-visual synchronized temporal jittering (Patrick et al., 2020).

### A.3 DOWNSTREAM EVALUATION

For evaluation on UCF101, we follow the test protocol of (Feichtenhofer et al., 2019): We sample 10 clips from each test video at a uniform time interval, and for each sampled clip, we take three equidistant spatial crops, resulting in a total of 30 views. We use each of the 30 views as input to our visual CNN and average the prediction scores from all 30 views to obtain the final prediction result. For evaluation on ESC-50 (Piczak, 2015a), we extract 10 equally spaced 2-second clips from each test audio sample. We use each of 10 clips as input to our audio CNN and average the prediction scores to obtain the final prediction result. For evaluation on Charades and Kinetics-Sounds, we use three audio-visual sequences with different spatial crops from a test video and max-pool/average the prediction scores from each sequence, respectively.

### A.4 OPTIMIZATION

In all experiments, we use the AMSGrad (Reddi et al., 2018) variant of AdamW (Loshchilov & Hutter, 2019) optimizer with $\beta_1 = 0.9$, $\beta_2 = 0.98$, L2 weight decay of 1e-4. We use a learning rate warm-up for the first 6% of iterations followed by a linear decay of learning rate.

From the observations of Lezcano-Casado and Martínez-Rubio (Lezcano-Casado & Martínez-Rubio, 2019), we have 10 times less learning rate for the orthogonal parameters than that for the non-orthogonal parameters: we use 1e-5 for the former and 1e-4 for the latter.

We pretrain our model on Kinetics-700 (Carreira et al., 2019) with a batch size 256 for 220K iterations and AudioSet (Gemmeke et al., 2017) with a batch size 300 for 220K iterations in the main experiments; for the ablation study, we use a much smaller batch size of 4 and pretrain our model on Kinetics-700 for 80K iterations.

For finetuning on UCF101, we train our model for 40K iterations with a batch size of 64 and learning rate of 0.02. For evaluation on ESC-50, we train a multi-class one-vs-all linear SVM on top of our fixed audio CNN for 38K iterations with a batch size of 128 and learning rate of 0.003. For finetuning on Charades, we train for 40K iterations with a batch size of 8, with learning rate of 0.001 for the classifier and CNN parameters, 1e-5 for the orthogonal parameters and 1e-4 for the rest parameters. For finetuning on Kinetics-Sounds, we train for 24K iterations with a batch size of 32, with learning rate of 0.005 for the classifier and CNN parameters, 1e-4 for the orthogonal parameters and 1e-3 for the rest parameters.

## B  EXTRA RESULTS FROM THE ABLATION STUDY

### B.1  NEGATIVE SAMPLING STRATEGIES

We proposed the content-aware negative sampling strategy (CANS) using pairwise $l_2$ distances between CNN embeddings. We introduced two variants of CANS: CANS-Dissimilar that favors negatives that are dissimilar to the positive instance and CANS-Similar that favors negatives that are similar to the positive instance. We chose to use CANS-Similar based on the results from our ablation study presented in the main paper, Table 1 (b).

Figure 3 (a) in this appendix provides additional evidence that supports our decision. We see that the loss of CANS-Disimilar initially drops rapidly but starts increasing around iteration 7K and continues to increase until around 15K; this is mainly caused by the visual MEP loss shown in Figure 3 (a-2). One explanation for this might that CANS-Disimilar is easier to solve than CANS-Similar, which causes the loss landscape of CANS-Disimilar to contain too many shallow local minima compared to that of CANS-Similar. Recall that we use a learning rate warm-up for the first 6% of iterations during pretraining; this roughly equals to the first 13K (out of 220K) iterations. Given this, we speculate that the model got out of a local minima around iteration 7K (most likely due to the increasing learning rate), and then eventually settled in another (bad) local minima after the warm-up period ended. Compared to this, we observe much milder learning dynamics with CANS-Similar: the loss decreases relatively slowly but steadily, and eventually leaps around 35K to go below the loss of CANS-Disimilar. We, again, believe that this is because CANS-Similar is more difficult to solve than CANS-Disimilar (as shown by the slower decrease in loss values), which caused the resulting loss landscape to contain steeper local minima. Our model eventually found one of those after round 40K of iterations, resulting in a better performing model in the downstream tasks shown in Table 1 (b) of the main paper (the loss kept slowly decreasing after iteration 50K).

### B.2  PARAMETER SHARING SCHEMES

We proposed a cross-Transformer weight sharing technique, which decomposes weight matrices with low-rank approximation. Recall that each layer of a Transformer contains the multi-head attention layer weights $\{W^q, W^k, W^v, W^b\}$ and the feed-forward layer weights $\{W^c, W^d\}$. We chose to share all six weight matrices across Transformers, though we could have shared any combination of them. To justify this design choice, we empirically compared three variants: (i) Multi-6 that do not share parameters across Transformers, (ii) Multi-6-Part_Att that shares only $\{W^q, W^k, W^v, W^b\}$ (but not $\{W^c, W^d\}$) and (iii) Multi-6-Part that shares all six weight matrices. Figure 3 (b) shows that there is not much difference between all the variants in terms of the loss curves; we chose to use Multi-6-Part that requires the least number of parameters. We showed that our approach outperforms Multi-6 in the ablation study (Table 1 (c-left) in the main paper).

### B.3  JUSTIFICATION FOR THE MEP LOSS FORMULATION

Since the multimodal Transformer $h_{AV}$ has access to both visual and audio inputs, one might think that the model could "leak" information about visual input into $\mathbf{z}^a$ and information about audio input into $\mathbf{z}^v$, which could make MEP trivial to solve. Here we show that this is not the case. By construction, we mask the same positions in audio and visual streams when designing the MEP task, so the model has no access to the masked input even in a cross-modal manner. Empirically, removing the third term in Eq. 6 ($\mathcal{L}_{\text{NCE}}([\mathbf{x}^a; \mathbf{x}^v], \tilde{\mathbf{z}})$) leads to performance degradation in Kinetics-Sounds, i.e., top-1 accuracy 66.7% vs. ours 67.5% (see Table 1), which suggests that solving the MEP task in the multimodal Transformer is beneficial to our model.

### B.4  JUSTIFICATION FOR THE CPP LOSS FORMULATION

Recall that our CPP loss has two terms; the first term uses the summary embeddings $\mathbf{s}_g$ and the second term uses output embeddings $\mathbf{s}_h$ sampled at random positions; see Eq. 7. One could argue that the two terms are redundant as bidirectional Transformers have "one-step" access to all the input embeddings, and thus solving CPP only with the summary embeddings (the first term) would be enough. This is not the case. We encode $\mathbf{s}_h$ with position-specific information through the time

embeddings $\mathbf{p}_t$, which makes every $\mathbf{s}_h$ different compared to $\mathbf{s}_g$. Empirically, we find that removing the second term of Eqn. 7 ($\mathbf{s}_h$) in our CPP loss leads to an inferior accuracy 66.9% vs. ours 67.5% on Kinetics-Sounds, suggesting its importance in learning.

### B.5 USE OF MODALITY EMBEDDINGS IN THE MULTIMODAL TRANSFORMER

We use modality embeddings $\mathbf{m}^v$ and $\mathbf{m}^a$ as part of input to the multimodal Transformer in order to distinguish embeddings coming from visual and audio Transformers. They are learnable weights trained end-to-end with other parameters. Conceptually, incorporating modality-discriminative embeddings is crucial because of our aggressive weight sharing scheme. Without them, the multimodal Transformer will see the output from audio/visual Transformers ($y^a$ and $y^v$) as if they are coming from the same distribution because the two Transformers share a large part of weights. Using modality embeddings encourages our model to preserve modality-specific information in the final output, and this empirically leads to performance improvements: ours 67.5% vs. without modality embeddings 67.1% on Kinetics-Sounds.

### B.6 ON THE IMPORTANCE OF END-TO-END PRETRAINING

Previous work in multimodal visual-and-language tasks (Tan & Bansal, 2019; Lu et al., 2019) point out that using partially fixed Transformers of different modalities is detrimental to multimodal representation learning (c.f., Sun et al. (2019a;b)). We make the same observation in our audio-visual learning scenario. We compare two variants of `Multi-6 Part` in Table 1 (c), each of which pretrains only the audio (or visual) CNN/Transformer in the first half of pretraining stage and then continues pretraining the remaining weights while fixing the weights of the audio (or visual) CNN/Transformer in the second half. This leads to inferior performance (audio-fixed 62.8% and visual-fixed 63.1% vs. ours 67.5%), which is consistent with the results reported in Tan & Bansal (2019); Lu et al. (2019).

