# OpenReview forum: "Parameter Efficient Multimodal Transformers for Video Representation Learning"
_ICLR.cc/2021/Conference — ICLR 2021 Poster_

### Official Review · AnonReviewer2 · 2020-10-28
**Official Blind Review #2**

**Rating:** 5
**Confidence:** 5

**Review:**

Summary:

In this paper, the authors propose a multimodal transformer network for audio-visual video representation learning.  To reduce parameters,  a new parameter sharing scheme is introduced. They further propose a Negative Sampling strategy to improve model training. Experiments are performed on several audio and video benchmarks.

Strengths:

(1) The proposed parameter sharing can reduce model sizes and preserve performance.

(2) The negative sampling strategy is content-aware and can improve data quality.

Weaknesses:

(1)  The technical novelty is limited. The main contributions are the proposed parameter sharing and negative sampling strategies. However, as discussed in the paper, parameter sharing across layers has already explored in previous works. In addition, improving negative sampling for audio-visual learning is also not a new idea. AVTS has already explored it. Thus, the contributions are pretty incremental.

(2) Transformer-like audio-visual networks have been used in recently published papers. For example, [1] uses an audio-visual transformer for audio event classification, and [2] proposes a new joint audio-visual transformer module with both self-attention and cross-modal attention.

[1] Boes, Wim, and Hugo Van hamme. "Audiovisual Transformer Architectures for Large-Scale Classification and Synchronization of Weakly Labeled Audio Events."ACM MM. 2019.

[2] Tian, Yapeng, Dingzeyu Li, and Chenliang Xu. "Unified Multisensory Perception: Weakly-Supervised Audio-Visual Video Parsing." ECCV (2020)

(3) Unfair comparison in Table 2(b). The authors validate the effectiveness of learned audio features on the ESC-50 dataset. Previous methods including SoundNet, L3, DMC, and AVTS extract features from pre-trained models and use an SVM to predict event categories without fine-tuning of audio networks.   However, the authors fine-tune the model on the ESC-50, which makes the comparison become unfair.

(4) With an additional multimodal transformer model, the proposed method fails to greatly improve performance. The previous methods such as AVTS and DMC, they only have an audio net and a visual net to compute loss functions and learn representations. However, the proposed model has a large additional multimodal transformer after slowfast visual net and ResNet-50 audio net. Even so, I fail to observe significant improvements over recent approaches.

*** Post-Rebuttal ***

Some of my concerns are addressed by the authors' rebuttal. The unfair experimental comparison has been fixed.

The proposed model indeed has some merits (e.g., parameter sharing and negative sampling). However, to me, the technical novelty of the paper is incremental.  In addition, with additional large transformer networks, the proposed model only achieves limited improvements over previous methods. The authors claim that the proposed model is more effective in handling long videos. But, only results on Charades and KS are shown without extensive comparisons.

Thus, I would like to keep my rating unchanged.

---

> ### Author Response · Authors · 2020-11-21
> **Thanks! We clarified novelty concerns and fixed mistakes on ESC-50 evaluation protocol**
>
> We appreciate your thoughtful and constructive feedback.
>
> * **Novelty:** Thanks for acknowledging that our main algorithmic contributions are the cross-modal/layer parameter sharing and content-based stochastic negative sampling strategies. However, the reviewer seems to be underwhelmed by both, possibly due to a misunderstanding. We would like to clarify our contributions:
>
>     * We made it clear that cross-layer sharing of Transformer weights has been done by Lan et al., 2020 (in fact, RNN is based on the same principle, where the notion of “layers” with shared weights is replaced by the steps to be unrolled in a recurrent manner). What we propose as new is the cross-modal sharing strategy, where we decompose weight matrices of the Transformer with a low-rank approximation. To the best of our knowledge, this hasn’t been done in the Transformer literature.
>
>     * AVTS [Korbar et al., 2018] does not perform content-based negative sampling -- they use time difference to approximate the similarly between clips. It is easy to see that such an assumption can easily be violated -- e.g., consider a workout video with repeated actions (push-ups), the time based approach will be ineffective in finding hard negatives because of the periodic actions.
>
> * **Audio-visual transformers:** Thanks for providing missing references; we have included those in our revised paper.
>
> * **ESC-50:** We appreciate the reviewer for pointing out this, which is our obvious mistake. We have now fixed this; see the updated results in Table 2. We have also clarified our experimental setting in Section A.4 (last paragraph) in the Appendix.
>
> * **Small gain on short duration clips:** We agree that the use of the Transformer does not provide significant gains on UCF-101 and ESC-50, which contains clips of relatively short duration (5 to 7 seconds). As R3 pointed out, one of the contributions in this work is that we aim for long-duration (30 seconds) video understanding, which has not been done before in the self-supervised learning literature. We believe that the previous work that use only audio/visual CNNs will remain to be competitive at short-duration tasks: those models can perform well by learning short-term statistical regularities rather than long-term dependencies in the data. Our focus is learning from long duration videos, and we believe that our results on short duration videos are promising given that the only supervision we provide to the CNNs is from the pretext tasks defined over long videos, which means the CNNs must learn from long-term dependencies rather then short-term regularities. We also report results on Charades that contain long videos; to the best of our knowledge, this is rarely done in the self-supervised video understanding literature.

---

### Official Review · AnonReviewer4 · 2020-10-28
**Useful contribution, well-written paper but only a few labs can replicate this**

**Rating:** 8
**Confidence:** 5

**Review:**

The paper is a nice read. It builds on a line of research on multi-modal video understanding that utilises transformers where these works: 1) fix one of the transformer models (e.g. BERT) and 2) utilise tokens and thus do not train the approach in an end-to-end fashion. This typical trend is due to the memory requirements for training a multi-modal transformer end-to-end.
To allow for end-to-end learning, the paper argues for shared parameters (across the network), primarily:
a) sharing weights in CNNs of the same model [understandable]
b) sharing weights between layers of transformer for the same modality
c) sharing weights between modality transformers
d) performing mid-fusion by having modality-specific transformer followed by cross-modality transformer,
e) sharing position encoding parameters between modalities and transformer layers
f) decomposing transformer weights, so some are distinct and others are shared
-- among other suggestions [I didn't list fully].
In addition to the above, the paper showcases the need for context-aware negatives, rather than random sampling, in line with a number of concurrent works that address this issue, some submitted to ICLR [which I reviewed as well coincidentally].

The paper then runs a number of experiments to prove their approach, all showcasing the combined advantage of end-to-end learning with shared parameters. This is tested on the "usual suspect" set of datasets: Kinetics, Audio Set, with downstream tasks on short-range datasets (e.g. UCF) as well as long-term (e.g. Charades). Performance improvement over the baseline is consistent.

Two aspects of the paper are disappointing,
First, the motivation that the approach will enable end-to-end learning with transformer, should be re-written to say: "This would enable end-to-end learning with multi-modal transformer for a handful of labs who have 64 V-100 GPUs which can be trained for 220K batches [I'm presuming that's many days/weeks]." It is quite impossible for almost all researchers to utilise the findings of this paper. It's true that the number of parameters has dropped significantly, but in any forward/backward pass, the memory requirements of multiple slow-fast (ResNet-50) with all transformers in memory and a necessarily large batch size keeps the same limitation of an "end-to-end-to-end" approach more likely to be used by the community. Apart from knowing of this finding, I am not sure how this will transform the community's go-to solutions.

Second, the experimental results (tables and commentary on tables) are not designed for easy consumption. This makes the readability of the experimental section below acceptable bar IMO. This should be fixable, but disappointing that it is submitted in the current form. Let me give you a few examples of how difficult it was to read the tables of results:
Ex1: Table 1, the caption talks about a, b and c but these are not referenced in the actual tables. It took me several minutes to realise you are referring to the two tables on the first row as the gap between the two tables is not easy to observe.
Ex2: Table 1, names of sampling methods "similar/dissimilar" do not align with how the paper describes hard negatives. Using varying terminology you need to rely on the brief description to know what's happening.
Ex3: Table 2, the decision to list the dataset references like this makes the table and checking the references an impossible task, many abbreviations (e.g. KS for Kinetics-Sounds) are not common. On the first two table within Table 2 you use Ours, which I presume is M-BERT in the right table? It is not clear why V- and A- were not tested independently for the tables on the left, but are ablated on Charades and Kinetics Sound.
These tables were very hard to follow/read and check for correctness.

A few minors:
*) I am not sure the results on Charade represent the SOA on this dataset. They seem to only reference the first 2017 paper as a baseline? The same for Kinetics-Sounds, this seems to be a very old baseline?
*) The manuscript talks about the audio being a "real valued audio signal" and it's only in the appendix that the log-mel-scaled spectrogram is explained. This can be deceiving to the reader.
*) Given very few can replicate these results, the fact that an input of 1 second was only tried in all experiments limits our knowledge of the impact of this critical parameter.
*) On the issue of Task 2 (correct pair prediction) other works have discussed the need for asynchronous understanding from the audio-visual signal that are worth referencing, e.g.
Kazakos and Zisserman (2019). Audio-Visual Temporal Binding for Egocentric Action Recognition. ICCV
Morgado et al (2020). Audio-Visual Instance Discrimination with Cross-Modal Agreement. ArXiv April 2020 [recent work understandably] https://arxiv.org/abs/2004.12943v1

---

> ### Author Response · Authors · 2020-11-21
> **Truly appreciate your overwhelmingly positive rating!**
>
> We truly appreciate your thoughtful and constructive comments (and an overwhelmingly positive rating!).
>
> * **Large GPU requirement:** This is unfortunately true, but the same challenge applies to most existing video understanding models (due to the heavy memory requirement of handling video data). Our work contributes to the video literature by reducing the model size. One important yet orthogonal direction is reducing the size of input data, e.g., by adaptively selecting which frames to process (e.g., AdaFrame [Wu et al., CVPR 2019]) or by learning directly from the compressed video format (e.g., CoViAR [Wu et al., CVPR 2018]). We believe that there is a long way to go for video learning; we hope our work moves the needle by contributing to efficient network modeling.
>
> * **Readability of Tables 1 and 2:** Overall, we had to squeeze the tables to allow for optimal use of space (the single-column format was challenging to put tables optimally). We have carefully incorporated your suggestions to improve the readability of the experimental results, including:
>
>     * Tab 1: We have added labels (a), … (d) in blue to visually highlight them and widened the gap between tables (thanks to R1’s tip on using \quad).
>
>     * Tab 1: We have revised the text to match the names of sampling methods; see the bottom of page 4 (CANS-Similar).
>
>     * Tab 2: We have clarified what models “Ours” refer to, i.e., we used V-CNN on UCF-101 and A-CNN on ESC-50 (both were taken from our full model that includes V/A/M-BERT).  Sorry for the heavy (and slightly inappropriate) use of abbreviations for the dataset names; we had to squeeze all the content for optimal use of space.
>
> * **Missing references:** Thanks! We have included the suggested references; see the first paragraph on page 4.
>
> * **SoTA on Charades and Kinetics-Sounds:** To our best knowledge, there is no reported baseline of self-supervised approaches on these datasets (only fully-supervised baselines exist). To provide extra comparison points, we have added V/A/M-CNN results for the Charades dataset.
>
> * **“real-valued audio-visual signals”:** We have added footnote 2 (page 4) to clarify that our inputs are RGB images and Mel-spectrograms. The statement itself is still valid, as our spectrogram contains real values (we treat them as a single channel image, following the usual practice in the computer vision literature).

---

### Official Review · AnonReviewer3 · 2020-10-29
**Well-motivated model design, difficult to evaluate importance of contribution**

**Rating:** 6
**Confidence:** 3

**Review:**

**Summary**
In this work, the authors present a method for learning audiovisual (AV) representations from videos using a Transformer-based model architecture. Since both video processing and Transformer-based model are memory-intensive, a parameter-reducing scheme is proposed, which facililates training the model end-to-end. The AV representations are learned by training the network to solve two self-supervised pertaining tasks, and subsequently evaluated on various audio/visual downstream tasks. An ablation analysis is performed to demonstrate the efficacy of the various contributions.

**Strengths**
- Leveraging unlabeled videos to learn powerful audiovisual representations is an important problem, and judging by the many recent papers on this topic, of much relevance to the community.
- The use of a Transformer-based model architecture is well-motivated, given its success on other multi-modal problems. To the best of my knowledge, videos of such long duration (30 sec.) have not yet been used in this setting, and the ability to process such long videos seems doable due to the contributions of this paper.
- The ablation study presented does a good job of highlighting the contribution of the design choices, and can be of importance to future works which seek to build upon this one.
- Empirical results on both audio and video understanding tasks demonstrate that the proposed method does indeed learn useful representations, and that multimodal training provides the expected boost to results.


**Concerns**
- The authors imply that using a partially fixed model (as has been done with multimodal vision/language tasks) is inferior to end-to-end training, hence the motivation for the proposed parameter-reducing technique, which is the major technical contribution of this work, as presented by the authors. This may very well be the case, but I would have liked to see evidence of that. Obviously, comparing fixed vs. end-to-end training of vision/language models using the proposed method would be very interesting to see, but is out of the scope of this work. However, perhaps comparing the proposed end-to-end audiovisual model with a similar fixed model would provide the insight necessary to determine the importance of your contribution.
- It seems to me that there are some previously reported self-supervised results missing from Table 2:
[1] obtain 93% accuracy on UCF101 and 85.8% on ESC using AudioSet for pretraining.
[2] obtain 91.3% on UCF101, and [3] 93.8% on UCF101, both using larger datasets.
This obviously a crowded space, with new results being published often, but I would have hoped to see larger performance gains on tasks that require more global reasoning, such as action recognition, given the ability of the proposed model to process very long sequences.
- Perhaps I didn't understand this correctly, but in Table 1, shouldn't CANS-dissimilar perform less well, since negative samples are "easy", therefore causing MEP to dominate?
- Nit: In equation (1), I think Q, K, and V might be missing the subscript 'i'?

**Additional comments**
- Abstract: The reader needs to read almost halfway through the abstract to get to what this work is about. IMO, either move a significantly reduced version of the "why" to after the "what", or perhaps just leave the motivation for the introduction.


[1] Alwassel, Humam, et al. "Self-supervised learning by cross-modal audio-video clustering." arXiv preprint arXiv:1911.12667 (2019).
[2] Miech, Antoine, et al. "End-to-end learning of visual representations from uncurated instructional videos." Proceedings of the IEEE/CVF Conference on Computer Vision and Pattern Recognition. 2020.
[3] Piergiovanni, A. J., Anelia Angelova, and Michael S. Ryoo. "Evolving Losses for Unsupervised Video Representation Learning." Proceedings of the IEEE/CVF Conference on Computer Vision and Pattern Recognition. 2020.

---

> ### Author Response · Authors · 2020-11-21
> **Thanks for thoughtful comments, we added extra results to help evaluate importance of contribution**
>
> We appreciate the positive comments and constructive feedback. We particularly thank the reviewer for pointing out that our work is the first to learn from videos of long duration (30 sec) in a self-supervised manner. This is indeed an important point to emphasize and poses great challenges that we address in this work.
>
> Based on the insightful comments, we have updated our paper with the following changes:
>
> * **Partially fixed model baseline:** We think this is a great suggestion! We have added a new section in the Appendix (B.6) showing the comparisons with those baselines for an extended discussion.
>
> * **Tab 2:** We’d be happy to discuss XDC [Alwassel et al., 2020] who also report AudioSet-pretrained results. We have also added two other references [2,3] as they are closely relevant to our work (though they used HowTo100M/YouTube-8M for pretraining). In general, we agree that video self-supervised learning is indeed a crowded space with new state-of-the-art results published often; performance on standard benchmarks such as UCF101/ESC50 are getting somewhat saturated, which contain short duration clips (5 to 7 seconds). We hope that our work will encourage the community to go beyond short-clip understanding and start tackling long video understanding.
>
> * **CANS-Dissimilar:** We believe that this is primarily due to two factors: 1) We discard instances that fall outside the 95% confidence interval, which helps remove ``"too easy" and `"too hard" negatives; those will be included in the two other baselines Current-Sequence and Current-MiniBatch; 2) We perform stochastic sampling using the embedding distance as the sampling probability. With a batch size of 256, there’s a good chance to include “low-probability” samples, e.g., hard negatives for CANS-Dissimilar.
>
> * **Eqn (1):** Thanks for spotting the missing subscript (we believe the reviewer meant “j”). We have fixed it.
>
> * **Abstract:** This is a great suggestion. We have shortened the “why” and extended the “what”.

---

### Official Review · AnonReviewer1 · 2020-11-01
**Sound paper with multiple ablations but lacking clarity**

**Rating:** 6
**Confidence:** 3

**Review:**

This paper studies modeling and training choices when designing a single model based on ConvNets and transformers for audio-visual representation learning. It proposes ablations for which weights/layers to share across modalities, when/where to fuse/join both modalities, and other modeling details (that matter). It also completes pre-training with 3 InfoNCE (audio-audio, visual-visual, audio-visual) with a binary classification loss about if two pairs of audio-visual are from the same or different videos. As strategies for negative sampling in audio and videos are different, it proposes to sample negatives that are similar in the ConvNets' embeddings. The models are pretrained on Kinetics-700 and AudioSet and evaluated on UCF101, ESC-50, and Kinectics-Sounds.

Contributions:
+ A sound, extensive set of ablations of the choices made along the way of modeling and training.
+ Good performance (overall best reported but for UCF101).

Limitations:
- The article is at times slightly difficult to follow. A few important pieces are mentioned too quickly: e.g. how exactly to do the content-aware negative sampling (bottom paragraph page 4), or how exactly to get UΣV from (maintain Σ orthogonal with a scale-squaring trick + first project Vx on the unit sphere). Some explanations come after the first introduction of the term.
- There is a small flaw in the parameters sharing experiments (Table 1, bottom) as the models that are compared have vastly different capacity, so maybe not the same training regimes. It would be more rigorous to also have an experiment where one reduces the (parameters) size of the model without sharing from 155M to ~34M (or 31M, and decompose W=UΣV for those weights too), and compare this to the all/part parameter shared models.
- Not a negative point in itself, but related to how this article is written: content-aware negative sampling is not new, see e.g. FaceNet (Schroff et al. 2015), On Mutual Information in Contrastive Learning for Visual Representations (Wu et al. 2020).

Overall, the paper presents multiple ablations and gives results that show that the model leverages both modalities, on challenging datasets. But it feels a bit too much like a tech report, where the most important bits of the contribution are given a too light algorithmic treatment (negative sampling, parameter sharing).

Nitpick: in Table 1 the left / right tabulars are way to close, put some \quad.

---

> ### Author Response · Authors · 2020-11-21
> **Appreciate your positive feedback. We improved clarity in the revision.**
>
> We appreciate your constructive feedback. We have carefully incorporated the suggestions into our revised version.
>
> * **Clarity:** We have extended our discussion of the content-aware negative sampling and the decomposition of $W$ into $U \Sigma V$.
>
> * **No “parameter shared” baseline (Tab 1):** Thanks for this great suggestion! We have added this baseline to Table 1(c) which results in 34M parameters (see Multi-2 which contains only 2 layers instead of 6). The inferior result of this baseline is consistent with the results in Table 1(d) and confirms the findings reported in the ALBERT paper [Lan et al., 2020], which showed that the depth of Transformers is critical to obtain powerful representations.
>
> * **Related work:** We have added the two references to our paper and discussed them; see the last paragraph in Section 4.
>
> * **Table 1 left/right tabulars:** Thanks for this tip! We have widened the gaps using \quad.

---

### Decision · Program_Chairs · 2021-01-07
**Final Decision**

**Decision:**

Accept (Poster)

**Comment:**

Three out of four reviewers are positive about the paper after the author response and during the discussion.

Strengths include
* The proposed method for parameter reduction in transformers allows end-2-end learning cross-modal representations especially on long videos, which has not been possible before
* Good performance on audio and video understanding
* Extensive set of ablations

Concerns include a somewhat incremental nature of the paper and the still large computational resources to run the experiments.
I think, both, the ideas and results are interesting to the community and recommend accept.